# Viral Metagenomic Analysis of the Fecal Samples in Domestic Dogs (*Canis lupus familiaris*)

**DOI:** 10.3390/v15030685

**Published:** 2023-03-06

**Authors:** Hongyan Wang, Zongjie Li, Chuanfeng Li, Yanfeng Ma, Qing Sun, Hailong Zhang, Guangbin Niu, Jianchao Wei, Huochun Yao, Zhiyong Ma

**Affiliations:** 1College of Veterinary Medicine, Nanjing Agriculture University, Nanjing 210095, China; 2Animal Medical Testing Center, Department of Animal Production, Faculty of Agricultural & Biological Engineering, Jinhua Polytechnic, Jinhua 321007, China; 3Shanghai Veterinary Research Institute, Chinese Academy of Agricultural Science, Shanghai 200241, China

**Keywords:** domestic dogs, canine diarrhea, gut virome, viral metagenomics, viral diversity, viral communities

## Abstract

Canine diarrhea is a common intestinal illness that is usually caused by viruses, bacteria, and parasites, and canine diarrhea may induce morbidity and mortality of domestic dogs if treated improperly. Recently, viral metagenomics was applied to investigate the signatures of the enteric virome in mammals. In this research, the characteristics of the gut virome in healthy dogs and dogs with diarrhea were analyzed and compared using viral metagenomics. The alpha diversity analysis indicated that the richness and diversity of the gut virome in the dogs with diarrhea were much higher than the healthy dogs, while the beta diversity analysis revealed that the gut virome of the two groups was quite different. At the family level, the predominant viruses in the canine gut virome were certified to be Microviridae, Parvoviridae, Siphoviridae, Inoviridae, Podoviridae, Myoviridae, and others. At the genus level, the predominant viruses in the canine gut virome were certified to be *Protoparvovirus*, *Inovirus*, *Chlamydiamicrovirus*, *Lambdavirus*, *Dependoparvovirus*, *Lightbulbvirus*, *Kostyavirus*, *Punavirus*, *Lederbergvirus*, *Fibrovirus*, *Peduovirus*, and others. However, the viral communities between the two groups differed significantly. The unique viral taxa identified in the healthy dogs group were *Chlamydiamicrovirus* and *Lightbulbvirus*, while the unique viral taxa identified in the dogs with diarrhea group were *Inovirus*, *Protoparvovirus*, *Lambdavirus*, *Dependoparvovirus*, *Kostyavirus*, *Punavirus*, and other viruses. Phylogenetic analysis based on the near-complete genome sequences showed that the CPV strains collected in this study together with other CPV Chinese isolates clustered into a separate branch, while the identified CAV-2 strain D5-8081 and AAV-5 strain AAV-D5 were both the first near-complete genome sequences in China. Moreover, the predicted bacterial hosts of phages were certified to be *Campylobacter*, *Escherichia*, *Salmonella*, *Pseudomonas*, *Acinetobacter*, *Moraxella*, *Mediterraneibacter*, and other commensal microbiota. In conclusion, the enteric virome of the healthy dogs group and the dogs with diarrhea group was investigated and compared using viral metagenomics, and the viral communities might influence canine health and disease by interacting with the commensal gut microbiome.

## 1. Introduction

There are approximately 380 trillion viruses that inhabit the body of humans and other mammals, which are termed the mammalian virome [1]. Until now, the viral communities of the mammalian virome have been largely unknown. However, the roles of the mammalian virome in shaping the commensal microbiome and their regulating effects on the body’s health and disease are relatively unexplored areas [2,3]. Compared to the eukaryotic viruses that directly inhabit mammalian organs and tissues, the prokaryotic viruses that inhabit the symbiotic bacterial communities should also be focused on. In fact, bacteriophages are the most abundant viruses in the mammalian virome and play a critical role in shaping the bacterial communities and maintaining the host’s intestinal homeostasis [4,5]. The temporal changes and spatial dynamics of the mammalian virome could be influenced by geography, ethnicity, diet, and other factors. Metagenomic analysis revealed that the viral diversities apparently vary according to the different inhabited body sites. For example, luminal samples from the domestic pig and rhesus macaque were mainly composed of the Caudoviricetes class, the Microviridae family, and other bacteriophages. However, the mucosal samples were mainly composed of the Astroviridae, Caliciviridae, and Parvoviridae families and other eukaryotic viruses [6]. Therefore, the mammalian virome is intimately linked to the host’s health and disease along the longitudinal axis of the gastrointestinal tract (GIT) and other organs.

The metagenomic sequencing technique and bioinformatics analysis have been utilized to detect and identify a large number of putative or unknown viruses from the vertebrate virome. Recently, the rapid development of metagenomics sequencing and analyzing technology has already characterized a vast array of novel phages that cannot be identified with the traditional culturing-based approaches [7]. Several evaluating strategies have been explored to optimize the amplifying, sequencing, and assembling methods for metavirome investigations, and metagenomic next-generation sequencing (mNGS) technology has been broadly applied for the clinical diagnostics of infectious diseases and the precise surveillance of newly emerged pathogenic viruses [8,9,10,11]. Benler et al. identified 3738 complete phage genomes which represented 451 putative genera and thousands of previously unknown phage taxa by searching the circular contigs that encoded phage hallmark genes in human gut metagenomes [12]. Koji Yahara et al. identified hundreds of viral contigs from human saliva using long-read metagenomic sequencing approaches, while 0–43.8% and 12.5–56.3% of these predicted phages and prophages were different from the previously reported oral phages [13]. The gut viral metagenomics analysis revealed that the perturbing gut virome and their interactions with the gut bacterial microbiome might be associated with necrotizing enterocolitis (NEC) onset in preterm infants [14]. When compared to the gut virome of healthy early-weaned piglets to piglets with diarrhea, 58 differential DNA viruses and 16 differential RNA viruses belonging to different taxonomic levels were identified, including 1 (family Dhakavirus) and 6 (phylum Artverviricota, class Revtraviricetes, order Ortervirales, family Retroviridae, genus Gammaretrovirus, and species Kirsten murine sarcoma virus) [15]. Moreover, a new strain of porcine enteric alphacoronavirus (PEAV GDS04 strain) was identified from suckling piglets with diarrhea using the shotgun metatranscriptome sequencing strategies [16]. Therefore, mNGS technology demonstrated prospective applications in clinical diagnostics, and the metagenomic sequencing technique might provide effective strategies to treat virus-induced infectious diseases in the future [17,18].

With the aim to prevent and control the transmission of potential epidemic viral diseases that originate from domestic dogs (*Canis lupus familiaris*), viral metagenomics could be used to detect the known or unknown viruses that inhabit domestic dogs’ bodies [19]. By scanning the virome of dogs with respiratory infections, many kinds of RNA and DNA viruses were identified, including canine parainfluenza virus (CPIV), canine respiratory coronavirus (CRCoV), carnivore bocaparvovirus (CBoV), canine circovirus (CanineCV), canine papillomavirus, and canine taupapillomaviruses [20]. Importantly, the current results detected using mNGS techniques demonstrated that the canine coronavirus had a high sequence similarity with the human coronavirus; therefore, the potential transmission risks of canine coronavirus to humans and other mammals should not be neglected [21,22]. Moreover, the coinfections of canine coronavirus with other pathogenic viruses (such as canine parvovirus, CPV) might induce virus variations and cause mild-to-severe clinical syndromes and might also accelerate the cross-species viral transmission between humans and companion dogs [23,24,25]. Therefore, studies of the gut virome in domestic dogs are meaningful for elucidating the epidemiological role of animal viruses in public health.

In this study, viral metagenomics was applied to investigate and analyze the signatures of the intestinal virome in healthy dogs and dogs with diarrhea.

## 2. Materials and Methods

### 2.1. Ethics Statement

The study design was approved by the Committee for Accreditation of Laboratory Animal Care and the Guideline for the Care and Use of Laboratory Animals of Shanghai Veterinary Research Institute, Chinese Academy of Agricultural Science (approval number: 20210616).

### 2.2. Animals and Sample Collection

The fecal samples of 50 domestic dogs were collected from pet hospitals located in the urban districts of Jinhua City, Zhejiang Province. The dogs were separated into two groups: a healthy dogs group (group H) and a dogs with diarrhea group (group D). All the fecal samples were collected in sterile cryotubes and stored at −80 °C for further analyses.

### 2.3. Nucleic Acid Extraction, Library Preparation, and Sequencing

The viral nucleic acids were extracted from viral-like particles using the MagPure Viral DNA/RNA Mini LQ Kit (R6662-02; Magen, Guangzhou, China) which was according to the manufacturer’s instructions. The qualities of the extracted viral nucleic acids were quantified with a spectrophotometry reader (NanoDrop, Thermo Fisher Scientific, Waltham, WA, USA) and 1.5% agarose electrophoresis. The amplified DNA was randomly sheared using ultrasound sonication and the produced fragments were collected using beads [26]. The sequencing libraries were constructed using a Nextera XT DNA sample preparation kit, and then the generated libraries were pooled and sequenced on an Illumina NovaSeq 6000 platform (Illumina, San Diego, CA, USA) with 150 bp paired-end reads from the Magigene Company (Guangzhou, China) [21].

### 2.4. Bioinformatics Analysis

The quality control of the sequenced raw reads was performed with Trimmomatic software to remove the low-quality data and acquire the clean data [27]. The filtered clean data were mapped to the reference genome of the domestic dog (*Canis lupus familiaris*) and the ribosomal database (silva.132) using SOAPaligner (v2.0.5, BGI, Shenzhen, China) and BWA (v0.7.17, BGI, Shenzhen, China) to avoid the confusion caused by the host [28,29]. The obtained high-quality reads were aligned and de novo assembled into contigs using IDBA (v 1.1.1, The University of Hong Kong, Hong Kong, China.) and SPAdes (v3.15.4, Russian Academy of Sciences, St. Petersburg, Russia) software [30,31]. The assembled long-length contigs were then classified by the GenBank nonredundant nucleotide (NT) database [32]. The phage–host prediction was inferred from the database of CRISPR spacers through a BLASTN query against the phage genomes, and the spacer was 95% identical over 95% of its length to a phage sequence were selected.

### 2.5. Phylogenetic Analysis

The virus sequences confirmed from the assembled contigs were aligned using CLUSTALW and were then trimmed to match the viral genomic regions [33]. The phylogenetic tree based on the near-complete genome sequences was constructed using the neighbor-joining method with the MEGA 6.0 software (Tokyo Metropolitan University, Tokyo, Japan.) package with 1000 bootstrap replicates and Kimura-2-parameter model [34].

## 3. Results

### 3.1. Overview of the Viral Metagenomics

To investigate the enteric virome of healthy dogs and dogs with diarrhea, a viral metagenomics analysis was carried out. By mixing every 5 fecal samples into a same sample pool, a total of 50 fecal samples were separated into 10 sample pools, which included 5 sample pools in the healthy dogs group and 5 sample pools in the dogs with diarrhea group. Using the Illumina NovaSeq platform, a total of 263,337,595 clean reads were filtered from 384,381,025 raw reads, and then 248,870,850 viral reads were obtained after 14,466,745 host reads were removed. Finally, all the identified high-quality reads were de novo assembled into 20,158 contigs and aligned against the viral protein database (Table 1).

### 3.2. The Alpha and Beta Diversity of Domestic Dogs’ Gut Virome

The Chao 1 and Simpson indexes were calculated to analyze the alpha diversity analysis of the domestic dogs’ intestinal virome, respectively. The compared results indicated that the richness and diversity of the intestinal virome in the dogs with diarrhea group were much higher than the healthy dogs group (Figure 1). Then, principal coordinate analysis (PCoA) was used for the beta diversity analysis of the domestic dogs’ enteric virome; the results demonstrated that the gut viral samples of the healthy dogs group and those of the dogs with diarrhea group were obviously separated into different clusters, respectively (shown in Figure 2).

### 3.3. Taxa Compositions of the Domestic Dogs’s Gut Virome

All the 20,158 assembled contigs were classified by the GenBank nonredundant nucleotide (NT) database; therein, 16,973 contigs were identified as DNA viruses, 8 contigs were identified as RNA viruses, and 3177 contigs were unclassified. Taxonomic analysis revealed that 81.39% of the assigned viral genomes were identified as phages, while other viruses accounted for 8.61% of the assigned viral genomes. At the family level, Microviridae, Parvoviridae, Siphoviridae, Inoviridae, Podoviridae, and Myoviridae were certified to be the predominant viruses in domestic dogs’ feces (Figure 3A). Moreover, *Protoparvovirus*, *Inovirus*, *Chlamydiamicrovirus*, *Lambdavirus*, *Dependoparvovirus*, *Lightbulbvirus*, *Kostyavirus*, *Punavirus*, *Lederbergvirus*, *Fibrovirus*, and *Peduovirus* were certified to be the predominant viruses at the genus level (Figure 3B). However, the specific viral communities in the healthy dogs group and the dogs with diarrhea group were obviously different.

### 3.4. Comparisons of the Gut Viral Communities in Healthy Dogs and Dogs with Diarrhea

The top 30 unique contigs in each sample pool were displayed at the genus level (shown in Figure 4). In the dogs with diarrhea group, the relative abundances of the pathogenic virus were much higher, such as *Protoparvovirus*, *Dependoparvovirus*, *Kostyavirus*, *Punavirus*, and other viruses. However, the relative abundances of phages in the healthy dogs group were obviously higher than those in the dogs with diarrhea group, including *Chlamydiamicrovirus*, *Lightbulbvirus*, *Spbetavirus*, and other viruses.

The different viral communities between the healthy dogs group and the dogs with diarrhea group were compared using the linear discriminant analysis effect size (LEfSe), and the predominant viruses with a linear discriminant analysis (LDA) score > 2 were identified. As shown in Figure 5, the unique viral taxa in the healthy dogs group were identified as *Chlamydiamicrovirus* and *Lightbulbvirus*. However, the unique viral taxa in the dogs with diarrhea group were certified to be *Inovirus*, *Protoparvovirus*, *Lambdavirus*, *Dependoparvovirus*, *Kostyavirus*, *Punavirus*, and other viruses.

### 3.5. Functional Annotations of the Identified Viral Genome

The functional annotations of the dog viral genome were performed by searching the Kyoto Encyclopedia of Genes and Genomes (KEGG) databases (https://www.kegg.jp/, accessed on 1 January 2023). As shown in Figure 6, a total of 45 pathways were predicted using the diamond software, including the replication and repair, nucleotide metabolism, amino acid metabolism, xenobiotics biodegradation and metabolism, carbohydrate metabolism, energy metabolism, folding, sorting and degradation, lipid metabolism, infectious disease: bacterial, and other pathways. The enrichment of KEGG pathways revealed the close relations between the viral functions and the complicated metabolism of their bacterial hosts.

### 3.6. Predictions of the Phages’ Bacterial Hosts

The phage–host prediction was performed using the database of CRISPR-spacers from prokaryotic genomes, and the potential spacer matches were generated with the SpacePHARER software [12]. Through querying the metagenomic phages, a total of 2784 dereplicated phage genomes which matched at least 1 CRISPR-Cas system were identified (Table 2). The predominant predicted bacterial hosts at the genus level were certified to be *Campylobacter* (1523), *Escherichia* (655), *Salmonella* (115), *Pseudomonas* (48), *Acinetobacter* (23), *Moraxella* (21), *Mediterraneibacter* (20), and others.

### 3.7. Canine Parvovirus

Phylogenetic analysis based on the near-complete genome sequences showed that the CPV strains collected in this study together with other CPV Chinese isolates clustered into a separate branch, which was distantly related to the CPV vaccine strains included in the CPV foreign isolates branch (shown in Figure 7).

### 3.8. Canine Adenovirus

Phylogenetic analysis based on the near-complete genome sequences showed that the CAV-2 strain D5-8081 belonged to the species Canine mastadenovirus A of Mastadenovirus and was closely related to the CAV-2 strains Toronto A26/61, 18Ra-54, and A2, which shared more than 98% identity at nt level (shown in Figure 8). Moreover, D5-8081 is the first near-complete genome sequence of CAV-2 from China to date.

### 3.9. Adeno-Associated Virus

Phylogenetic analysis based on the near-complete genome sequences showed that the AAV-5 strain AAV-D5 belonged to the species Adeno-associated dependoparvovirus B of the genera Dependoparvovirus and was closely related to the AAV-5 strains AAV-5(Y18065), AAV-5(AF085716), AAV-Go.1(DQ335246), and HeB-NA1(OM451155), sharing more than 92% nt identity (shown in Figure 9). Moreover, AAV-D5 is also the first near-complete genome sequence of AAV-5 from dogs to date.

## 4. Discussion

The mammalian gut contains an enormous quantity of bacterial viruses; however, the roles of bacteriophages or phages in shaping the gut microbiome and influencing the host’s health have still been rarely investigated [2]. Several previous studies demonstrated that the viral diversities in the gut virome of different mammalian individuals apparently varied; therefore, there was an urgent need to identify and classify the phages and other viruses in the mammalian gut virome [12,35]. Canine diarrhea is a common intestinal illness which could be caused by viruses, bacteria, parasites, and other pathogens (Figure 10). Viral metagenomics analysis could be applied as a powerful tool to explore new and existing viruses, including canine distemper virus, canine parvovirus type 2, canine amdoparvovirus, the rotavirus, canine herpesvirus, canine papillomavirus, canine bufavirus, canine influenza virus, and canine parainfluenza virus [36,37,38]. Moreover, viral metagenomics could also be used to investigate the pathogens of chronic enteropathy, idiopathic meningoencephalomyelitis, and other diseases [39,40]. In particular, the potential transmission risks of canine viruses to humans and other mammals should also be focused on. Therefore, viral metagenomics analysis might play an essential role in the fields of veterinary medicine for studying the canine gut virome, and might also provide important information for evaluating the roles of animal viruses in public health.

In the current study, 50 fecal samples from healthy dogs and dogs with diarrhea were collected and then mixed into 10 sample pools. By filtering out the low-quality reads and removing the contaminations caused by ribosomes and host sequences, a total of 248,870,850 viral reads were obtained and were then assembled into 20,158 contigs (Table 1). The Chao 1 and Simpson indexes were, respectively, calculated for the alpha diversity analysis, and the results indicated that the richness and diversity of the gut virome in the dogs with diarrhea group were much higher than the healthy dogs group (Figure 1). In fact, in the condition of canine diarrhea, the intestinal immune function was disrupted by the invaded pathogens. The virus-eradicating ability of dogs with diarrhea was lower than the healthy dogs; therefore, the richness and diversity of the gut virome in the dogs with diarrhea group were much higher than the healthy dogs group. The beta diversities of the two groups were analyzed using principal coordinate analysis (PCoA), and the results demonstrated that the gut viral samples of the two groups were obviously different (Figure 2). Therefore, the beta diversity of the canine enteric virome was also changed by the diarrhea.

The viral taxonomic analysis revealed that most of the identified viral genomes were certified to be phages (81.39%), and other viruses only accounted for 8.61% of the assigned viral genomes. The predominant viruses at the family level were identified as Microviridae, Parvoviridae, Siphoviridae, Inoviridae, Podoviridae, Myoviridae, and others (Figure 3A). At the genus level, the predominant viruses were identified as *Protoparvovirus*, *Inovirus*, *Chlamydiamicrovirus*, *Lambdavirus*, *Dependoparvovirus*, *Lightbulbvirus*, *Kostyavirus*, *Punavirus*, *Lederbergvirus*, *Fibrovirus*, *Peduovirus*, and others (Figure 3B). In the condition of diarrhea, the dog’s gut immune system function was altered and the balance of the gut ecosystem was disrupted. Therefore, the viral communities in the healthy dogs group and the dogs with diarrhea group apparently differed.

The top 30 unique contigs in each sample were displayed at the genus level (shown in Figure 4). The different viral communities between the two groups were compared using the linear discriminant analysis effect size (LEfSe), and the predominant viruses with a linear discriminant analysis (LDA) score >2 were shown (Figure 5). The unique viral taxa in the healthy dogs group were identified as *Chlamydiamicrovirus* and *Lightbulbvirus*. However, the unique viral taxa in the dogs with diarrhea group were identified as *Inovirus*, *Protoparvovirus*, *Lambdavirus*, *Dependoparvovirus*, *Kostyavirus*, *Punavirus*, and other viruses. Therefore, the different viral communities in the healthy dogs group and the dogs with diarrhea group might play different biological roles. From searching the Kyoto Encyclopedia of Genes and Genomes (KEGG) databases, the predicted functions of the viral genome are shown in Figure 6. The enrichment of KEGG pathways revealed the close relations between the viral genome functions and the metabolism pathways of the bacterial hosts.

The phage–host prediction was performed using the CRISPR-spacers from prokaryotic genomes; a total of 2784 dereplicated phage genomes which matched at least 1 CRISPR-Cas system were identified (Table 2). At the genus level, the predominant predicted bacterial hosts were *Campylobacter*, *Escherichia*, *Salmonella*, *Pseudomonas*, *Acinetobacter*, *Moraxella*, *Mediterraneibacter*, and others. In fact, the commensal bacterial communities played a crucial role in regulating the host’s immune system functions, and could also help the host to synthesize vitamins and provide energy supply. Therefore, the gut virome might influence the canine gut environment by shaping the complex ecosystem of the commensal gut microbiome.

Through frequent contact with domestic dogs, the infected virus and other pathogens might have the risk of transferring to humans and inducing zoonoses [21]. Several contagious viruses (including canine parainfluenza virus, canine respiratory coronavirus, carnivore bocaparvovirus, canine circovirus, canine papillomavirus, and canine taupapillomaviruses) were certified to be able to cause infections and affect canine health. In the current study, different types of viral genomes were identified by searching the NCBI database. Canine parvovirus type 2 (CPV-2) belongs to the Parvoviridae family, Parvovirinae subfamily, *Protoparvovirus* genus, and Carnivore Protoparvovirus 1 species, and CPV-2 is a worldwide-distributed virus which was considered as the major viral cause of dog gastroenteritis [25,34]. Phylogenetic analysis based on the near-complete genome sequences revealed that five CPV strains were identified (Figure 7). The five CPV strains which were identified in this study together with other CPV Chinese isolates clustered into a separate branch, which was distantly related to the CPV vaccine strains in the CPV foreign isolates branch. Canine adenovirus (CAV) is also widely distributed and can be classified into canine adenovirus type 1 (CAV-1) and canine adenovirus type 2 (CAV-2). The infection of CAV could cause severe harm to domestic dogs, while CAV-1 could induce the infectious canine hepatitis (ICH) and CAV-2 could cause the infectious tracheobronchitis (ITB) [41]. The CAV-2 strain D5-8081 which was identified in this study belonged to the species Canine mastadenovirus A of Mastadenovirus and was closely related to the CAV-2 strains Toronto A26/61, 18Ra-54, and A2, which shared more than 98% identity at the nt level (Figure 8). Until now, D5-8081 was the first near-complete genome sequence of CAV-2 from China. Adeno-associated virus (AAV) is broadly used as a viral delivery system for gene therapy, especially for various forms of the blood-clotting disorder hemophilia. However, dogs treated with AAV gene therapy were identified with clonal expansions of transduced liver cells which demonstrated potential genotoxicity for liver tumors [42]. The AAV-5 strain AAV-D5 which was identified in this study belonged to the species Adeno-associated dependoparvovirus B of the genera *Dependoparvovirus* and was closely related to the AAV-5 strains AAV-5(Y18065), AAV-5(AF085716), AAV-Go.1(DQ335246), and HeB-NA1(OM451155), sharing more than 92% nt identity (Figure 9). Moreover, AAV-D5 was also the first near-complete genome sequence of AAV-5 from dogs up to now. Therefore, the coinfections of CPV-2, CAV-2, and AAV might affect the host’s gut health and cause canine diarrhea.

## 5. Conclusions

In all, the enteric virome of the healthy dogs group and the dogs with diarrhea group was investigated and compared using viral metagenomics. The alpha diversity analysis indicated that the richness and diversity of the gut virome in the dogs with diarrhea were much higher than the healthy dogs, while the beta diversity analysis revealed that the signatures of the gut virome in the two groups were quite different. Moreover, the viral communities between the two groups also differed significantly. Therefore, the enteric virome might influence canine health and disease by interacting with the commensal gut microbiome. Investigations on the enteric virome of domestic dogs could explore new and existing viruses and might also provide important information for evaluating the roles of gut viruses in public health.

## Figures and Tables

**Figure 1 viruses-15-00685-f001:**
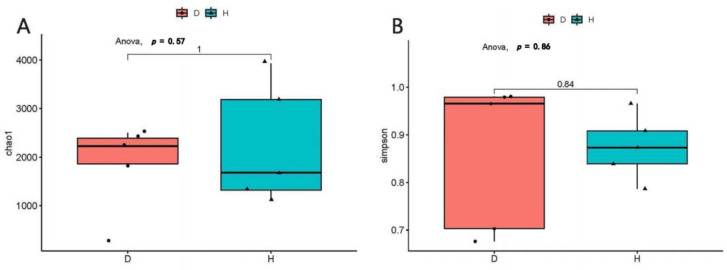
The alpha diversity analyses of intestinal virome in the healthy dogs and the dogs with diarrhea. At the genus level, the Chao1 index (**A**) and the Simpson index (**B**) were calculated, respectively.

**Figure 2 viruses-15-00685-f002:**
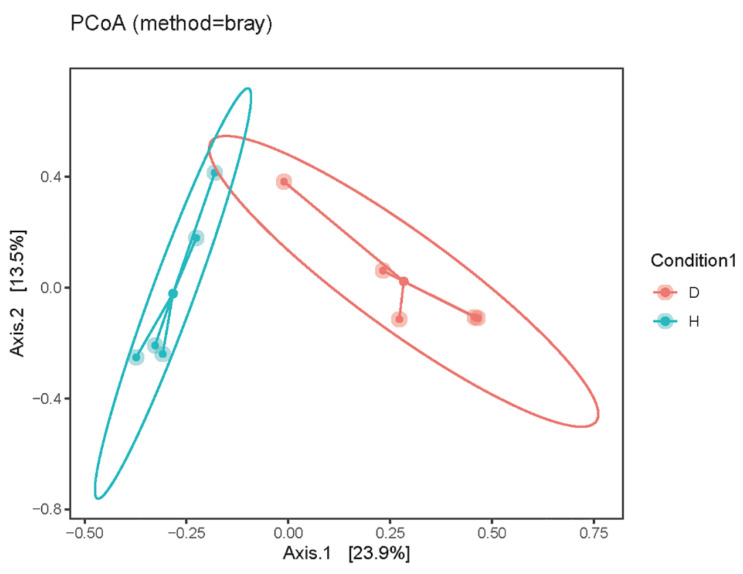
The beta diversity of intestinal virome in the healthy dogs and the dogs with diarrhea were analyzed using principal coordinate analysis (PCoA).

**Figure 3 viruses-15-00685-f003:**
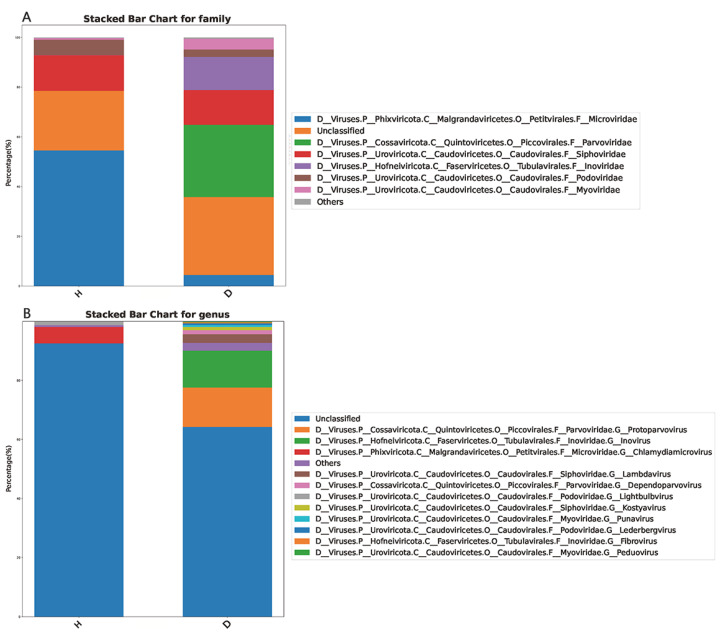
The compositions of intestinal virome in the healthy dogs and the dogs with diarrhea at the family (**A**) and genus class (**B**) levels.

**Figure 4 viruses-15-00685-f004:**
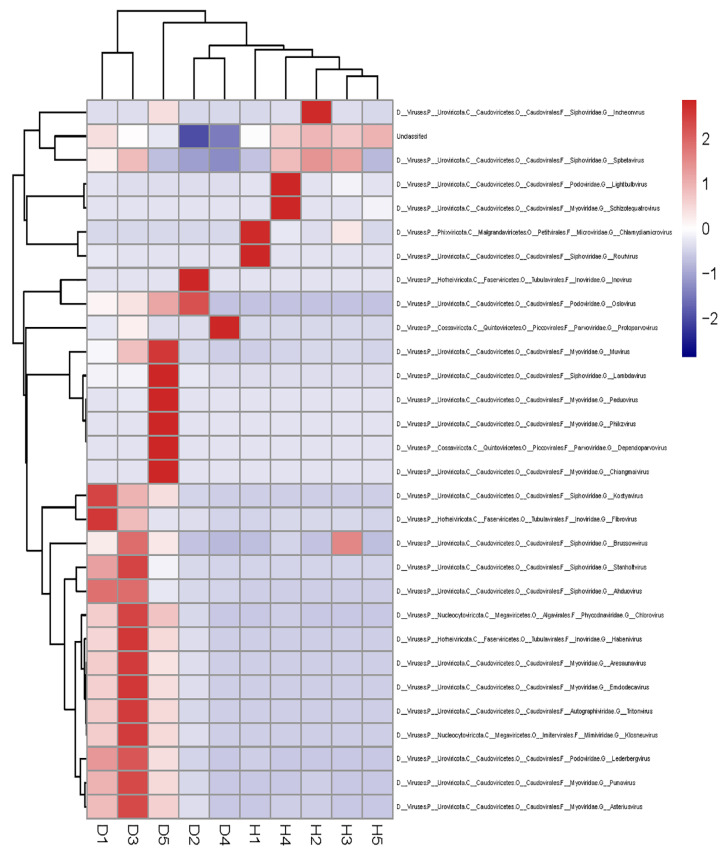
Heatmap of contigs with the top 30 most abundant sequence reads in each sample. The boxes colored from blue to red demonstrate the abundance of virus reads aligned to each contig, and the virus genus is provided in the right text column.

**Figure 5 viruses-15-00685-f005:**
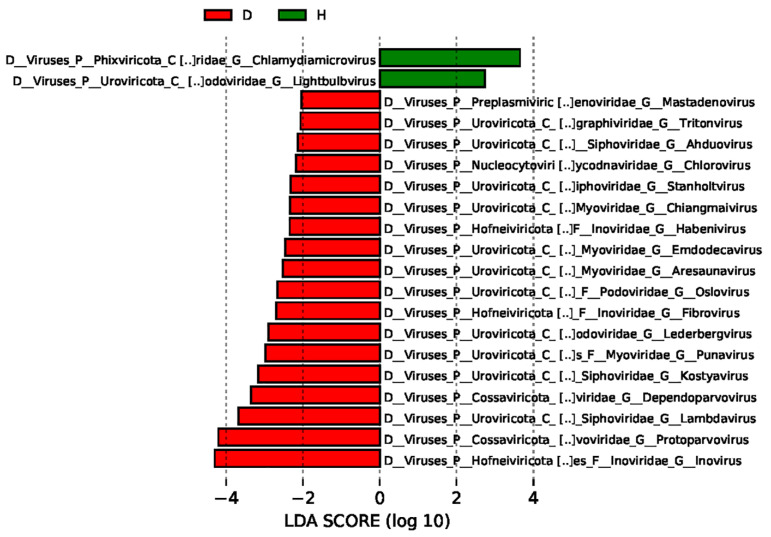
The linear discriminant analysis effect size (LEfSe) of intestinal virome in the healthy dogs and the dogs with diarrhea. LEfSe was performed to search for the taxa in which the relative abundances were significantly different between the two groups, and the LDA Score was calculated and is shown.

**Figure 6 viruses-15-00685-f006:**
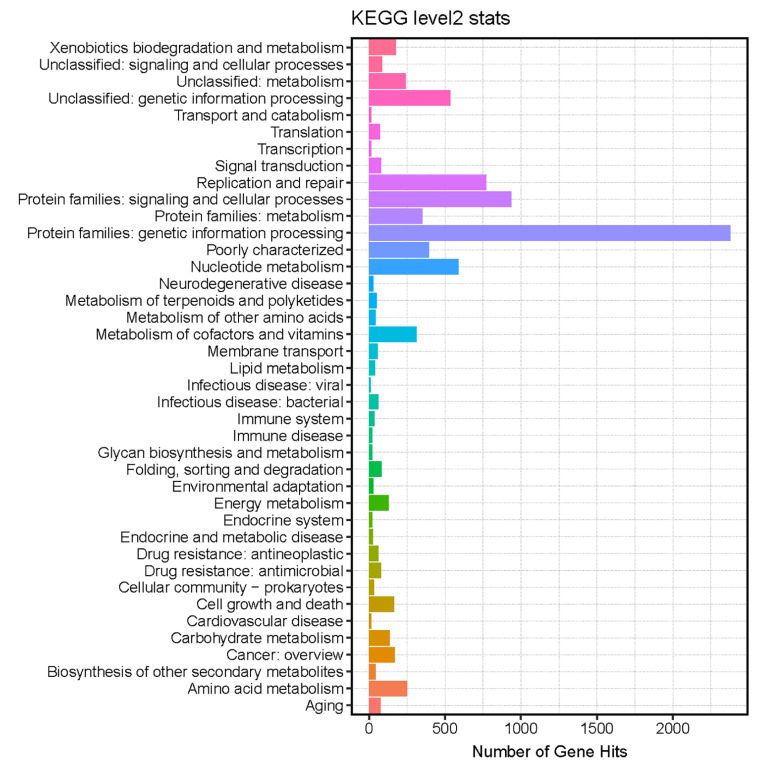
The gene annotations of KEGG function. The pathways related to the complicated metabolism and gastrointestinal diseases were identified.

**Figure 7 viruses-15-00685-f007:**
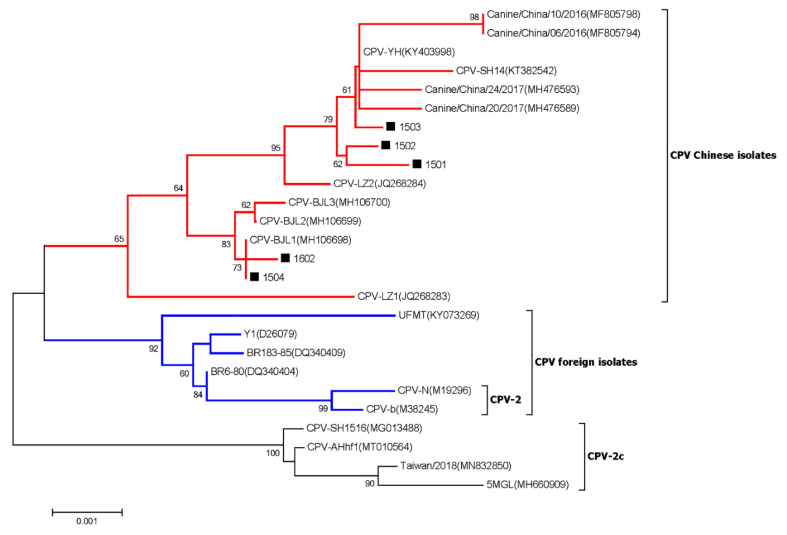
Phylogenetic analysis based on the near-complete genome sequences of CPV strains collected in this study and other representative strains retrieved from GenBank. The scale bar indicates nucleotide substitutions per site. CPV strains collected in this study are labeled with a black solid square (■). GenBank accession numbers of other CPV strains are indicated following the virus strains in the branches.

**Figure 8 viruses-15-00685-f008:**
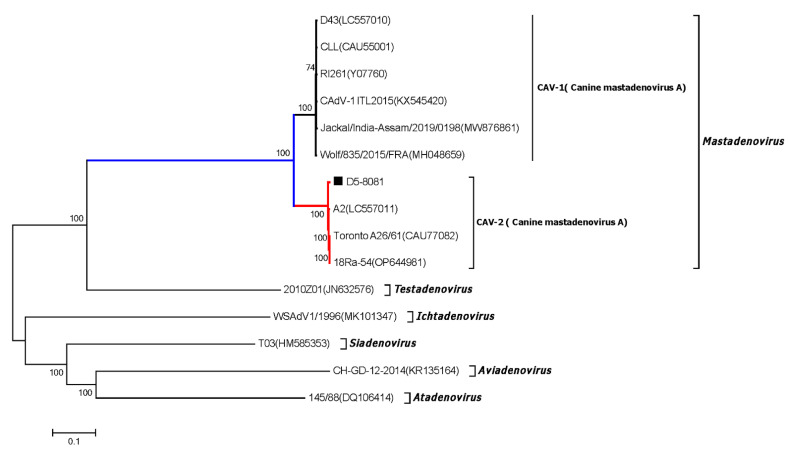
Phylogenetic analysis based on the near-complete genome sequences of CAV-2 strain D5-8081 collected in this study and other representative strains retrieved from GenBank. CAV-2 strain D5-8081 collected in this study are labeled with a black solid square (■). GenBank accession numbers of other adenovirus representative strains in six genera are indicated following virus strains in the branches.

**Figure 9 viruses-15-00685-f009:**
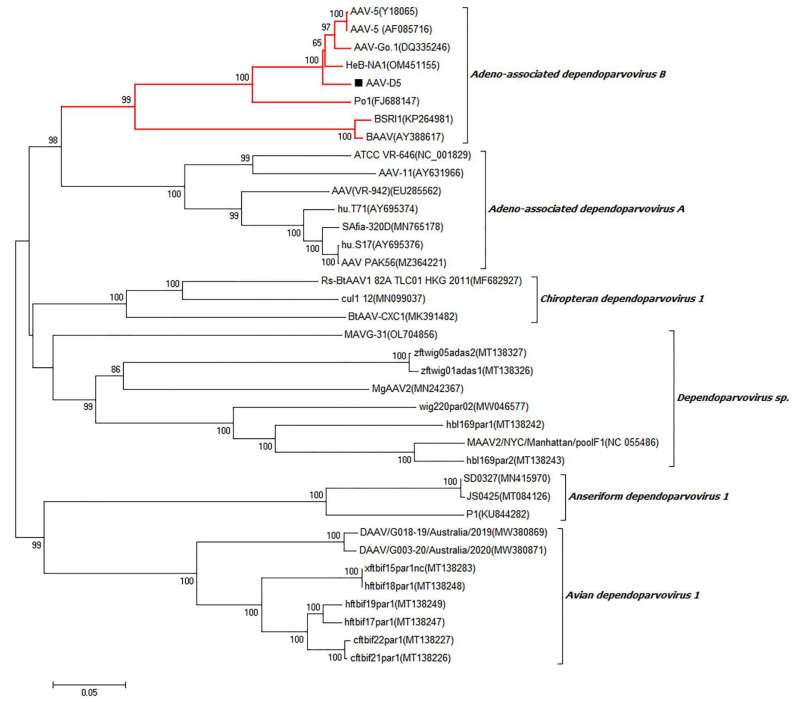
Phylogenetic analysis based on the near-complete genome sequences of AAV-5 strain AAV-D5 collected in this study and other representative strains retrieved from GenBank. AAV-5 strain AAV-D5 collected in this study is labeled with a black solid square (■). GenBank accession numbers of other AAV representative strains in the genera Dependoparvovirus are indicated following virus strains in the branches.

**Figure 10 viruses-15-00685-f010:**
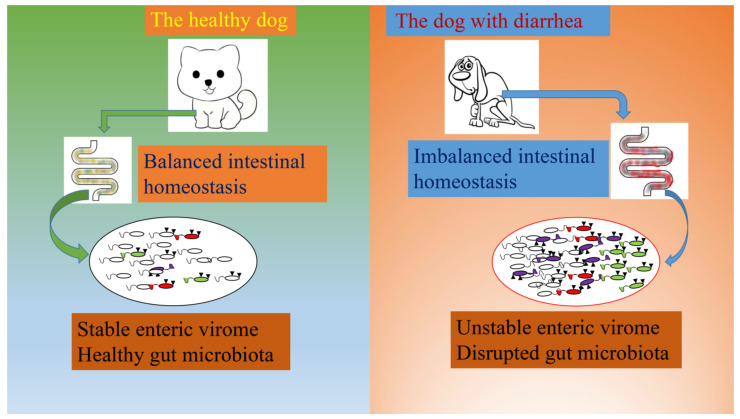
Schematic diagram of the canine enteric virome. The complicated interactions among the enteric virome, gut microbiota, and the domestic dogs were showed.

**Table 1 viruses-15-00685-t001:** The quality control of sequenced data and the assembled contigs.

Raw Reads	Clean Reads	*Canis lupus* Familiaris Reads	Surplus Reads	Total Contigs
384,381,025	263,337,595	14,466,745	248,870,850	20,158

**Table 2 viruses-15-00685-t002:** The predicted viral hosts at the genus level.

The Predicted Host Taxa of the Phages	Count
P__Proteobacteria.C__Epsilonproteobacteria.O__Campylobacterales.F__Campylobacteraceae.G__Campylobacter	1523
P__Proteobacteria.C__Gammaproteobacteria.O__Enterobacterales.F__Enterobacteriaceae.G__Escherichia	655
P__Proteobacteria.C__Gammaproteobacteria.O__Enterobacterales.F__Enterobacteriaceae.G__Salmonella	115
P__Proteobacteria.C__Gammaproteobacteria.O__Pseudomonadales.F__Pseudomonadaceae.G__Pseudomonas	48
P__Proteobacteria.C__Gammaproteobacteria.O__Pseudomonadales.F__Moraxellaceae.G__Acinetobacter	23
P__Proteobacteria.C__Gammaproteobacteria.O__Pseudomonadales.F__Moraxellaceae.G__Moraxella	21
P__Firmicutes.C__Clostridia.O__Eubacteriales.F__Lachnospiraceae.G__Mediterraneibacter	20
P__Firmicutes.C__Bacilli.O__Bacillales.F__Listeriaceae.G__Listeria	14
P__Firmicutes.C__Clostridia.O__Eubacteriales.F__Clostridiaceae.G__Clostridium	14
P__Proteobacteria.C__Gammaproteobacteria.O__Enterobacterales.F__Enterobacteriaceae.G__Klebsiella	14
P__Fusobacteria.C__Fusobacteriia.O__Fusobacteriales.F__Fusobacteriaceae.G__Fusobacterium	11
P__Firmicutes.C__Bacilli.O__Lactobacillales.F__Streptococcaceae.G__Streptococcus	8
P__Proteobacteria.C__Gammaproteobacteria.O__Xanthomonadales.F__Xanthomonadaceae.G__Xanthomonas	8
P__Proteobacteria.C__Betaproteobacteria.O__Neisseriales.F__Neisseriaceae.G__Neisseria	7
P__Actinobacteria.C__Actinomycetia.O__Bifidobacteriales.F__Bifidobacteriaceae.G__Bifidobacterium	6
P__Actinobacteria.C__Actinomycetia.O__Streptomycetales.F__Streptomycetaceae.G__Streptomyces	6
P__Firmicutes.C__Bacilli.O__Bacillales.F__Staphylococcaceae.G__Staphylococcus	6
P__Firmicutes.C__Bacilli.O__Lactobacillales.F__Enterococcaceae.G__Enterococcus	6
P__Firmicutes.C__Bacilli.O__Lactobacillales.F__Lactobacillaceae.G__Lacticaseibacillus	6
P__Firmicutes.C__Clostridia.O__Eubacteriales.F__Oscillospiraceae.G__Faecalibacterium	6

## Data Availability

The data supporting this study are openly available on the NCBI sequence read archive (SRA) under Bio Project: PRJNA917074, PRJNA917095, PRJNA917097, PRJNA917099, PRJNA917103, PRJNA917104, PRJNA917113, PRJNA917115, and PRJNA917118, PRJNA917120.

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
