# Peer review of "Viral Metagenomic Analysis of the Fecal Samples in Domestic Dogs (Canis lupus familiaris)"

_viruses, 2023, doi:10.3390/v15030685_

Round 1

Reviewer 1 Report

Canine diarrhea is a common intestinal illness. Wang et al reported the enteric viral communities of the healthy dogs and the dogs with diarrhea. The alpha diversity analysis and beta diversity were analyzed and found that the richness and diversity of the gut virome in the dogs were quite different. Overall, this is an interesting concept. Here are my comments:

1. No insights that would justify the rationale and necessity for examining the fecal samples in domestic dogs are provided. Any previous observations prompted the authors to examine the virome of dogs? A more thorough and logical presentation is required for readers to understand the paper rationale and why fecal samples of dogs were singled out. This is especially important.

2. In general, the author’s conclusions are sound, but the representations of the data in some figures are not up to the modern standards in terms of what is acceptable.

3. Page 3, before merging all the qualified sequences, the barcodes or primers were removed or not? How the effective sequences were clustered into operational taxonomic units? using a ??% threshold? Any redundancies were removed?

4. It would be easier for readers to understand the paper if the authors could provide a schematic diagram outlining the main findings of the study.

5. The conclusion seems out of place in the context of the data presented. It needs more explanation if the authors choose to take a generalist approach.

Author Response

Response to the reviewer's Comments

Dear Reviewer,

Too much thanks for the expert comments concerning our manuscript. Those comments are all valuable and very helpful for revising and improving our paper. We have studied comments carefully and have made corresponding corrections as required. The responses to the reviewer's comments are listed as following:

Canine diarrhea is a common intestinal illness. Wang et al reported the enteric viral communities of the healthy dogs and the dogs with diarrhea. The alpha diversity analysis and beta diversity were analyzed and found that the richness and diversity of the gut virome in the dogs were quite different. Overall, this is an interesting concept. Here are my comments:

  1. No insights that would justify the rationale and necessity for examining the fecal samples in domestic dogsare provided. Any previous observations prompted the authors to examine the virome of dogs? A more thorough and logical presentation is required for readers to understand the paper rationale and why fecal samples of dogswere singled out. This is especially important.

 Response1:Several previous studies (Shi, Ying et al. 2021, Frontiers in veterinary science vol. 8 695088. 7; Tao, Shiyu et al., Microbiology spectrum vol. 10,4 (2022): e0169822; Gong, Lang et al., Emerging infectious diseases vol. 23,9 (2017): 1607–1609) had used the fecal samples to study the mammalian gut virome. Our previous work also used the fecal samples to investigate the gut microbiota (Li, Zongjie et al.2022, Frontiers in microbiology vol. 13 872230). Therefore, it is rationale to examine the fecal samples for the enteric virome research in domestic dogs.

  1. 2. In general, the author’s conclusions are sound, butthe representations of the data in some figures are not up to the modern standards in terms of what is acceptable.

 Response2:The figure legends had been revised as required.

  1. Page 3, before merging all the qualified sequences,the barcodes or primers were removed or not?How the effective sequences were clustered into operational taxonomic units? using a ??% threshold? Any redundancies were removed?

  Response3:In the present study, the sequencing libraries were directly constructed using a Nextera XT DNA sample preparation kit, therefore libraries generating strategies were different from the 16S rRNA gene amplicons. Given that the raw data that were obtained by sequencing always include a certainproportion of low-quality reads, the Trimmomatic was used here to remove the paired reads as follows; (i) with adapter, (ii) those containing low-quality base (sQ 20) over 20%,(iii) those arising from PCR duplications, as well as (iv) those with a polyX sequence toimprove the accuracy of reads for follow-up analyses.

  1. It would be easier for readers to understand the paper if the authors could provide a schematic diagram outlining the main findings of the study.

   Response4: The schematic diagram had been added.

  1. The conclusion seems out of place in the context of the data presented. It needs more explanation if the authors choose to take a generalist approach.

 Response5: The conclusion had been changed.

Reviewer 2 Report

The authors analyzed and compared the gut virome in healthy dogs and dogs with diarrhea. The  diversity analysis revealed that the gut virome of the two groups were quite different. Moreover, the viral communities between the two groups also differed significantly. Therefore, the enteric virome might influence the canine health and disease by interacting with the commensal gut microbiome. Several questions should be answered by the authors as below.

1. The front of the virus name is different, for example, the front is different between Line 23 and Line 25. Please check the following similar situations.

2. So many spelling mistakes. such as, Line 214: The word included should be including. 

3. The annotation of Figure 6 should be more detailed.

4. Line 246 &247, Line 259 &260: How did you come to the conclusion? If all the genome sequence of CAV-2 or AAV-5 strains in the GeneBank have been aligned?

5. How did you get the sequences used for phylogenetic analysis of canine parvovirus, canine adenovirus, and Adeno-associated virus?

6. Why did you choose the three viruses to do the phylogenetic analysis ?

Author Response

Response to the reviewer's Comments

Dear Reviewer,

Too much thanks for the expert comments concerning our manuscript. Those comments are all valuable and very helpful for revising and improving our paper. We have studied comments carefully and have made corresponding corrections as required. The responses to the reviewer's comments are listed as following:

The authors analyzed and compared the gut virome in healthy dogs and dogs with diarrhea. The  diversity analysis revealed that the gut virome of the two groups were quite different. Moreover, the viral communities between the two groups also differed significantly. Therefore, the enteric virome might influence the canine health and disease by interacting with the commensal gut microbiome. Several questions should be answered by the authors as below.

  1. The front of the virus name is different, for example, the front is different between Line 23 and Line 25. Please check the following similar situations.

Response1:   The front of the virus name has been changed.

  1. So many spelling mistakes. such as, Line 214: The word included should be including. 

Response2: All the spelling mistakes in the context has been corrected.

  1. The annotation of Figure 6 should be more detailed.

Response3:   The annotation of Figure 6 has been changed.

  1. Line 246 &247, Line 259 &260: How did you come to the conclusion? If all the genome sequence of CAV-2 or AAV-5 strains in the GeneBank have been aligned?

Response4: The complete genome sequences of CAV-2 have been determined previously, however, now the existing three complete genome sequences of Toronto A26/61, A2 and 18Ra-54 were from Canada, Japan and South Korea, respectively. Although the complete genome sequences of AA-5 from human and animals have been determined previously, the complete genome sequence of AAV-5 in China were from himalayan marmot, not dog.  Therefore, D5-8081 and AAV-D5 are considered as the first near-complete genome sequences separately in China. In addition, all the genome sequences of CAV-2 or AAV-5 strains in the GeneBank have been aligned in our manuscript.

  1. How did you get the sequences used for phylogenetic analysis of canine parvovirus, canine adenovirus, and Adeno-associated virus?

Response5: All genomic sequences used for phylogenetic analysis of canine parvovirus, canine adenovirus, and Adeno-associated virus were retrieved from GenBank. The relevant illustration has been described in the corresponding figure captions.

  1. Why did you choose the three viruses to do the phylogenetic analysis ?

Response6: Canine parvovirus and canine adenovirus are important causative agents of viral diarrhea of dogs. Moreover, they were only found in the dogs of the diarrhea group, not the healthy group, indicating they may be the main pathogens causing diarrhea in dogs in this study. Genetic evolutionary analysis of these two viruses contributed to understand the viral evolutionary characteristics and genotyping, which can provide a reference for developing treatments and vaccine. Although adeno-associated virus was apathogenic, it can be developed into a vector carrying foreign genes for gene therapy. Phylogenetic analysis of dog-origin AAV assisted in the study of genetic evolutionary characteristics and analyzing the potential as an effective viral vector. 

Round 2

Reviewer 1 Report

No further comments

Reviewer 2 Report

no